# OFELIA: Optical Flow-based Electrode LocalIzAtion

**Xinyi Wang**[1,2,3]                                                        XINYIWANG@MAIL.USTC.EDU.CN
[1] *School of Biomedical Engineering, Division of Life Sciences and Medicine, University of Science and Technology of China, Hefei, Anhui, 230026, P.R.China*
[2] *Suzhou Institute for Advanced Research, University of Science and Technology of China, Suzhou, Jiangsu, 215123, P.R.China*
[3] *Shanghai MicroPort EP MedTech Co., Ltd. Shanghai, 201318, P.R.China*

**Zikang Xu**[1,2]                                                          ZIKANGXU@MAIL.USTC.EDU.CN

**Qingsong Yao**[4]                                                     YAOQINGSONG19@MAILS.UCAS.EDU.CN
[4] *Key Lab of Intelligent Information Processing of Chinese Academy of Sciences (CAS), Institute of Computing Technology, CAS, Beijing 100190, China*

**Yiyong Sun**[3]                                                           YIYONG.SUN@EVERPACE.COM

**S.Kevin Zhou**[1,2,4*]                                                     SKEVINZHOU@USTC.EDU.CN

**Editors:** Accepted for publication at MIDL 2024

## Abstract

Catheter ablation is one of the most common cardiac ablation procedures for atrial fibrillation, which is mainly based on catheters with electrodes collecting electrophysiology signals. Catheter electrode localization facilitates intraoperative catheter positioning, surgical planning, and other applications such as 3D model reconstruction. In this paper, we propose a novel deep network for automatic electrode localization in an X-ray sequence, which integrates spatiotemporal features between adjacent frames, aided by optical flow maps. To improve the utility and robustness of the proposed method, we first design a saturation-based optical flow dataset construction pipeline, then finetune the optical flow estimation to obtain more realistic and contrasting optical flow maps for electrode localization. The extensive results on clinical-challenging test sequences reveal the effectiveness of our method, with a mean radial error (MRE) of 0.95 mm for radiofrequency catheters and an MRE of 0.71 mm for coronary sinus catheters, outperforming several state-of-the-art landmark detection methods.

**Keywords:** Catheter Electrode Detection, Optical Flow

## 1. Introduction

Atrial fibrillation (AFib), atrial flutter, and premature ventricular contractions are prevalent manifestations of cardiac arrhythmias. Frequent cardiac arrhythmias may give rise to serious consequences, for instance, AFib can lead to blood clots in the heart (Staerk et al., 2017). Compared with pharmaceutical interventions, catheter-based radiofrequency ablation techniques in cardiac electrophysiology (EP) stand as the standard surgical intervention for the definitive treatment of rapid cardiac arrhythmias, characterized by immediate therapeutic effects and high success rates (Mark et al., 2019; Parameswaran et al., 2021). The electrode

---

* Corresponding Author

is one of the most crucial components of the catheters, which is used for EP signal collection and catheter localization. Catheter electrode localization can facilitate intraoperative catheter positioning, surgical planning, 3D model reconstruction and so on. However, due to the unstable imaging quality and the intersections among multiple catheters throughout the clinical surgical procedure, it is hard for physicians to locate the electrodes precisely in real-time X-ray images. Thus, it is necessary to develop accurate catheter placement detection methods not only to alleviate burdens for clinicians in the surgery but help novice doctors get familiar with this surgery.

There are research works (Ambrosini et al., 2017; Yang et al., 2019; Nguyen et al., 2020) that formulate this task as a segmentation task, which locates the catheters by the center of the mask of electrodes. Other studies adopt single frame landmark detection (LD) methods to solve the problem. Catheter segmentation information is indeed helpful, but the labeling is time-consuming. Experimentally, we observe that the optical flow map can to some extent provide shape and boundary information of catheter electrode without the need for specific annotations (Demoustier et al., 2023). Besides, optical flow maps can provide temporal context in X-ray videos, which could make full use of *correlation between successive frames* and the *label-free shape context*.

To solve the electrode localization problem and motivated by the the above empirical findings, we go beyond a single frame and propose an effective and easy-to-implement network, **O**ptical **F**low-based **E**lectrode **L**ocal**I**z**A**tion (**OFELIA**) for electrode localization in an X-ray video sequence.

Specifically, we introduce the optical flow map between consecutive frames as the input to the LD network, which not only presents the position changes over time but also provides estimated shape information of the electrodes (As shown in Fig. 3). Besides, as the ground truth of optical flow is difficult to acquire in our task, we construct a simulated dataset, the Flying-Catheter Dataset, based on several pre-trained RAFT (Teed and Deng, 2020) models, to train the optical flow estimator.

This paper offers the following contributions:

1. We propose an OFELIA network, which integrates the spatiotemporal information in an X-ray sequence for precise electrode localization. To the best of our knowledge, it is the first to introduce optical flow into electrode localization in an X-ray sequence;

2. To bridge the gap between natural images and X-ray images, we construct a Flying-Catheter dataset and fine-tune RAFT for accurate optical flow estimation.

3. Extensive experiments on test datasets illustrate that the OFELIA method outperforms the state-of-the-art electrod detection methods on two commonly used catheters.

## 2. Related Works

**Optical Flows in MedIA.** Optical flow maps are occasionally used in medical image areas. The FW-Net (Nguyen et al., 2020) introduces an end-to-end framework, which combines a segmentation network, an optical flow network, and a flow-guided warping function to learn temporal continuity for real-time catheter segmentation in a 2D X-ray fluoroscopy sequence. Optical flow maps are also utilized in (Xue et al., 2022) to achieve echocardiography segmentation. FlowReg (Mocanu et al., 2021) introduces a two-part deep learning

system for unsupervised neuroimaging registration, combining 3D affine adjustments and 2D deformable fine-tuning based on the optical flow network to enhance global and local alignment of medical imaging volumes.

**Single-Image Landmark Detection.** In (Yao et al., 2020), a multi-task U-Net is implemented to predict both heatmap and offset maps of landmarks simultaneously. In (McCouat and Voiculescu, 2022), an efficient contour-hugging landmark detection method with uncertainty estimation is depicted. In (Zhu et al., 2022), a light-weighted universal anatomical landmark detection model has been developed.

**Video Landmark Detection.** Compared to single-image landmark detection, video landmark detection utilizes the information between frames. In (Ullah et al., 2019), a tracker is implemented to extract the tip detection results in the last frame as a reference for segmenting the tip in successive frames. U-LanD (Jafari et al., 2022) capitalizes on the uncertainty inherent in landmark prediction to achieve automatic detection of landmarks in key frames of videos. The most similar work to ours is ConTrack (Demoustier et al., 2023), which uses both spatial and temporal context for tip detection and tracking. It incorporates multiple template frames and a search frame for catheter segmentation and initial tip detection. Subsequently, it uses successive segmentation to refine tips with optical flow maps. However, it relies on catheter segmentation masks, necessitating extensive annotations. In contrast to this, our OFELIA only requires point annotations, which is much easier to obtain.

## 3. Method

**Problem Definition** Let $D = \{(X_t, Y_t)\}_{t=1}^{N}$ represents an X-ray video sequence with $N$ frames, where $X_t \in \mathbb{R}^{w \times h}$ is the $t$-th video frame with a shape of (w, h), and $Y_t \in \mathbb{R}^{N_e \times w \times h}$ denotes the position of $N_e$ landmarks in frame $t$. Specifically, suppose that the $k$-th landmark of the $t$-th frame is at $(x, y)$. $Y_t^k$ is defined:

$$Y_t^k(i, j) = \begin{cases} 1, \text{ if } i = x \ \& \ j = y; \\ 0 \text{ , otherwise.} \end{cases} \tag{1}$$

OFELIA aims to train a network $f(\cdot)$, which takes the $\{X_t\}_{t=1}^{N}$ as input and predicts the locations of electrodes in each frame, i.e., $\{\hat{Y}_t\}_{t=1}^{N}$.

**OFELIA Network** The architecture of OFELIA is shown in Fig. 1, which aims to predict the landmark positions $Y_t$ using both $X_t$ and $X_{t+1}$. Particularly, we try to solve this problem by utilizing the information between the two frames, i.e., the optical flow map.

Optical flow, a concept to measure the motion of objects in continuous images, is widely used for tracking cells in fluoroscopy (Guo et al., 2013). By computing the direction and magnitude of the velocity, the optical flow map can be used to describe the temporal-spatial information of the electrodes in an X-ray video.

Specifically, we first predict the optical flow map $F_{t \to t+1}$ between $X_t$ and $X_{t+1}$ using an optical flow estimator $\varphi^*(\cdot)$, which takes two frames as input and predicts the pixel movement between them, i.e., $F_{t \to t+1} = \varphi^*(X_t, X_{t+1})$. Then, the raw image $X_t$ and optical flow map $F_{t \to t+1}$ are concatenated and sent to a modified U-Net (Ronneberger et al., 2015). The encoder of the U-Net is a pre-trained ResNet-34, while the decoder consists of 5 upsampling layers with 512, 256, 128, 64, and 32 channels respectively. As there are $N_e$

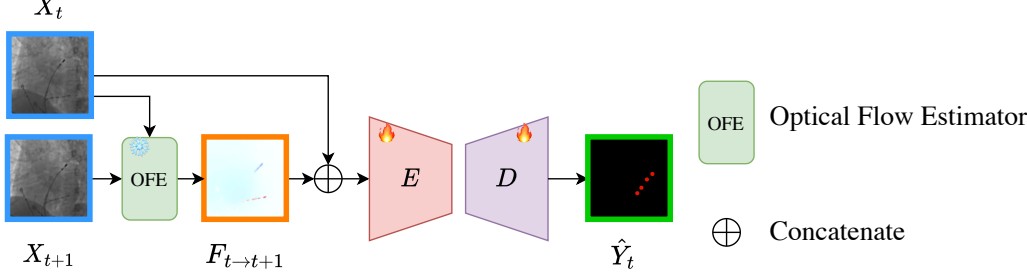

Figure 1: Overview of OFELIA. During the training procedure, the optical flow estimator is frozen and only the parameters of the Encoder $E$ and the Decoder $D$ are updated.

electrodes that need to be localized in each frame, we add a convolution layer after the last layer of the decoder to squash the number of channels to $N_e$. The $n$-th channel represents the predicted localization probability map of the $n$-th electrode and the position with the highest probability is regarded as the final prediction. The loss function $\mathcal{L}$ is defined as the average channel-wise cross-entropy loss between the predicted probability map $\hat{Y}_t^{N_e}$ and ground truth $Y_t^{N_e}$, given as $\mathcal{L} = \frac{1}{N_e} \sum_{k=1}^{N_e} \mathcal{L}_{CE}(\hat{Y}_t^k, Y_t^k)$.

The information captured by the optical flow map illustrates the catheter movement along with time and provides additional spatial shape information of the electrodes, which usually requires manual annotations. Combining X-ray images and optical flow maps can drive the neural network to pay more attention to the electrode part of the field of view. This contributes to the precise localization of the electrodes.

**Optical Flow Estimator** Estimating the optical flow map is essential for the final prediction of our OFELIA. However, the ground truth optical flow is inaccessible in our task. Besides, most applicable optical flow estimators are trained on natural image datasets, and the large domain gap results in poor prediction. Thus, we **simultaneously** construct a simulated X-ray optical flow dataset (called Flying-Catheter) and train a task-specific flow estimator on it. The pipeline is shown in Algorithm. 1.

First, we adopt four publicly available pre-trained optical flow estimators on our catheter dataset, including *raft-chairs*, *raft-kitti*, *raft-sintel* and *raft-things* which are variations of RAFT (Teed and Deng, 2020) trained different natural RGB datasets. For each frame, $X_t$, the four estimators predict four flows, $\hat{F}_{t \to t+1}^p, p \in \{c, k, s, t\}$. Then, we propose a saturation-channel-based selection algorithm to decide the final pseudo optical flow $\hat{F}_{t \to t+1}^*$ for each frame. Specifically, the predicted optical flow map, which is coded in the RGB format following RAFT, is first converted to the HSV format. We empirically find that the saturation channel in HSV is good at partially capturing the eletrode boudaries. Then the mean saturation factor (SF) of each map is calculated using the below:

$$\mathrm{SF}^p = \frac{1}{N_e} \sum_{i=1}^{N_e} \hat{F}_{t \to t+1}^p(x_i, y_i), \tag{2}$$

---

**Algorithm 1 Optical Flow Estimator**

---

**Input:** Original Dataset: $D_{ori} = \{X_t\}_{t=1}^{N}$, Pre-trained RAFTs: $\varphi^c, \varphi^k, \varphi^s, \varphi^t$, Original RAFT: $\varphi$, Quality Control Threshold: $\alpha$, Flying-Catheter Dataset: $D_{f-c} = \varnothing$

  $t \leftarrow 1$

  **repeat**

    Predict Optical Flow between $X_t$ and $X_{t+1}$: $\hat{F}_{t \to t+1}^{p} \leftarrow \varphi^p(X_t, X_{t+1}), p \in \{c, k, s, t\}$;

    Convert $\hat{F}_{t \to t+1}^{p}$ into HSV color space;

    Compute Saturation Factor $SF^p$ for each flow map using Eq. (2);

    Find the optical flow map with the highest SF using Eq. (3);

    **if** $SF^{p^*} >= \alpha$ **then**

      Add sample pair to the Flying-Catheter Dataset: $D_{f-c} = D_{f-c} \cup \{X_t, X_{t+1}, \hat{F}_{t \to t+1}^{p^*}\}$;

    **end if**

  **until** $t = N - 1$

  Update $\varphi^*$ on $D_{f-c}$ using gradient descent;

**Output:** Fine-tuned RAFT model: $\varphi^*$

---

where $(x_i, y_i)$ is the coordinate of the $i$-th electrode landmark in frame $X_t$. The pseudo optical flow map is defined as the predicted optical flow map with the highest SF:

$$\hat{F}_{t \to t+1}^{*} = \hat{F}_{t \to t+1}^{p^*}, \quad p^* = \arg\max_{p} SF^p. \tag{3}$$

However, due to the large gap between natural images and X-ray images, even the best of the four predictions may have low quality. Thus, we conduct a quality control procedure on the constructed dataset by discarding samples with an SF smaller than a threshold of $\alpha$. Finally, we finetune the original RAFT on the remaining dataset, denoted as the Flying-Catheter Dataset, for task-specific optical flow estimation. Compared to the original RAFT and RAFT trained on other natural image datasets, RAFT trained on the Flying-Catheter dataset can better capture spatial information of the electrodes, which serves as a strong prior knowledge for landmark detection using OFELIA (as shown in Fig. 3).

## 4. Experiments and Results

**Experiment Settings**

Dataset. This study uses an in-house multi-center dataset of fluoroscopic sequences captured during cardiac ablation procedures and animal experiments. Most of the frames include two types of commonly used catheters, Coronary Sinus (CS) and Radio-Frequency (RF) catheters. All the landmarks are defined as the center point of the electrodes except for the first landmark of the RF catheter, which is defined as the tip of the RF catheter. This results in 14 landmarks (4 for RF and 10 for CS) in each frame. The dataset is annotated by two skilled engineers using LabelMe (Russell et al., 2008) and reviewed by three professional clinical experts. The training and test sets consist of 560 sequences(14,768 frames) and 346 sequences(7,711 frames), respectively. To evaluate the stabilization and generalization of our proposed method, we extract two clinical-challenging (CCA) subsets, which consist of frames of a special scene in the operation (53 sequences, 575 frames, denoted as Test-DSA Subset) and frames where the catheters are partially obstructed (145 sequences, 2,266

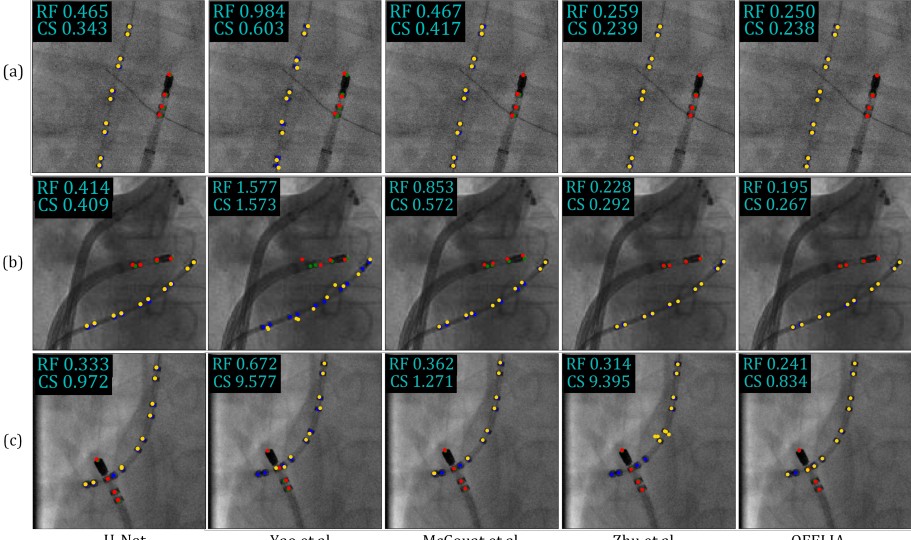

Figure 2: Qualitative results on the Test set(a), Test-DSA subset(b) and Test-OBS subset(c). The ground truth and predicted electrodes of CS catheter are in Blue and Yellow, respectively, and those of RF catheter are in Green and Red, respectively.

frames, denoted as Test-OBS Subset). These two test sets are more difficult for catheter electrode detection as they involve more complex situations.

Metrics We use mean radial error (MRE) to measure the Euclidean distance between prediction and ground truth. Additionally, the successful detection rate (SDR) is assessed across three different radii: 1mm, 2mm, and 4mm.

Implementation details. Our model is implemented in PyTorch and trained on an NVIDIA A100 GPU. The image pairs are augmented by random rotation, intensity scaling, and elastically deformation, and resized to $640 \times 640$ before being sent to the network. The network training is conducted utilizing the Adam optimizer, commencing with a learning rate of 0.001 and employing a batch size of 4 for 20 epochs. Learning rate adjustments are implemented by decreasing it by a factor of 0.1 at epochs 4, 8, 12, and 16. The threshold for quality control of the Flying-Catheter Dataset is set to $\alpha = 0.5$.

**Main Results**
We compare OFELIA with several commonly used algorithms for medical landmark detection (Ronneberger et al., 2015; Yao et al., 2020; McCouat and Voiculescu, 2022; Zhu et al., 2022), and the quantitative results are shown in Table 1. We observe that OFELIA outperforms the baseline methods on most of the metrics on the test sets. This might result from the temporal information introduced from the flow map, as other methods focus on spatial features only. Besides, our OFELIA presents good generalization on the CCA sequences as the SDR drop is much lower compared to other methods, which is justifiable as bringing in extra knowledge improves the robustness of the network and provides an aid to deal with difficult situations. We also present qualitative results of different detection methods in Fig. 2, where the MRE of each image is on the top of the image. Our method outperforms other methods significantly. More results are in the Appendix.

Table 1: Results on the Test sets and CCA subsets. **Best** and Second Best are highlighted.

| | **Test Dataset** | | | | | | | |
|---|---|---|---|---|---|---|---|---|
| | **RF Catheter** | | | | **CS Catheter** | | | |
| **Model** | MRE±STD ↓ | SDR (%)↑ | | | MRE±STD ↓ | SDR (%)↑ | | |
| | (mm) | 1mm | 2mm | 4mm | (mm) | 1mm | 2mm | 4mm |
| U-Net* | 1.59±4.91 | 84.97 | 91.62 | 94.51 | 1.08±4.84 | 92.20 | 95.66 | 97.11 |
| Yao et al.* | 3.42±8.99 | 78.32 | 83.53 | 86.42 | 2.28±12.65 | 81.79 | 87.57 | 94.22 |
| McCouat et al.* | 1.46±3.35 | 82.66 | 91.62 | 93.64 | 1.06±3.07 | 88.73 | 96.53 | 97.98 |
| Zhu et al.* | 1.29±3.19 | 86.13 | 92.20 | 94.51 | 0.93±2.85 | 93.06 | 97.69 | 98.27 |
| OFELIA (Ours) | **0.95±2.02** | **90.17** | **95.38** | **96.82** | **0.71±1.75** | **95.66** | **98.27** | **99.42** |
| | **Test-DSA Subset** | | | | | | | |
| | **RF Catheter** | | | | **CS Catheter** | | | |
| **Model** | MRE±STD ↓ | SDR (%)↑ | | | MRE±STD ↓ | SDR (%)↑ | | |
| | (mm) | 1mm | 2mm | 4mm | (mm) | 1mm | 2mm | 4mm |
| U-Net* | 3.86±9.09 | 77.36 | 79.25 | 83.02 | 0.86±1.20 | 83.02 | 94.34 | 96.23 |
| Yao et al.* | 6.27±12.51 | 64.15 | 69.81 | 75.47 | 5.10±10.53 | 66.04 | 71.70 | 75.47 |
| McCouat et al.* | 2.65±6.88 | 81.13 | 84.91 | 88.68 | 0.66±0.28 | 88.68 | 94.34 | 100.00 |
| Zhu et al.* | 2.10±5.29 | 83.02 | 86.79 | 88.68 | **0.52±0.27** | 94.34 | 100.00 | 100.00 |
| OFELIA (Ours) | **1.52±3.30** | **86.79** | **88.68** | **94.34** | 0.64±0.19 | **96.23** | 100.00 | 100.00 |
| | **Test-OBS Subset** | | | | | | | |
| | **RF Catheter** | | | | **CS Catheter** | | | |
| **Model** | MRE±STD ↓ | SDR (%)↑ | | | MRE±STD ↓ | SDR (%)↑ | | |
| | (mm) | 1mm | 2mm | 4mm | (mm) | 1mm | 2mm | 4mm |
| U-Net* | 2.85±6.42 | 73.10 | 81.38 | 84.83 | 1.28±2.87 | 78.62 | 88.28 | 92.41 |
| Yao et al.* | 4.43±10.09 | 71.72 | 78.62 | 81.38 | 3.35±18.33 | 74.48 | 82.07 | 91.72 |
| McCouat et al.* | 1.82±4.35 | 82.07 | 87.59 | 90.34 | 0.85±1.15 | 86.21 | 91.03 | 95.17 |
| Zhu et al.* | 1.80±4.29 | 83.45 | 89.66 | 91.03 | 1.27±3.75 | 88.97 | 93.79 | 96.55 |
| OFELIA (Ours) | **1.58±2.82** | **86.21** | **91.03** | **93.10** | **0.73±1.53** | **91.72** | **95.86** | **97.24** |

\* Implemented with the official code.

## Abalation Study

To evaluate the efficiency of our proposed method, we conduct several ablation studies and present the results below and in the Appendix.

RAFT trained on Flying-Catheter. We use four pre-trained RAFT and FlyingCath RAFT, to predict the optical flow map, and the result is shown in Fig. 3. From Fig. 3 we can find that the prediction of FlyingCath RAFT contains more spatial information of the electrodes, and the boundary is clearer, which brings a strong prior knowledge for landmark detection.

The introduction of optical flow. The proposed OFELIA takes frame $X_t$ and the corresponding optical flow $F_{t \to t+1}$ as input. Here we replace $F_{t \to t+1}$ with (1) The segmentation component of $X_t$ with highest probability from Segment Anything Model (SAM (Kirillov et al., 2023)) without prompt; (2) The subsequent frame $X_{t+1}$; (3) Estimated optical flow map using RAFT trained on natural dataset $F_{t \to t+1}^s$; (4) OFELIA without extra information. The result in Table 2 illustrates that, although aggregating extra information can improve the utility of landmark detection, the usage of optical flow tends to be more efficient. Besides, using optical flow maps generated by Flying-Cath RAFT exhibits better performance than the original RAFT model, which also proves the necessity of fine-tuning RAFT on the constructed Dataset.

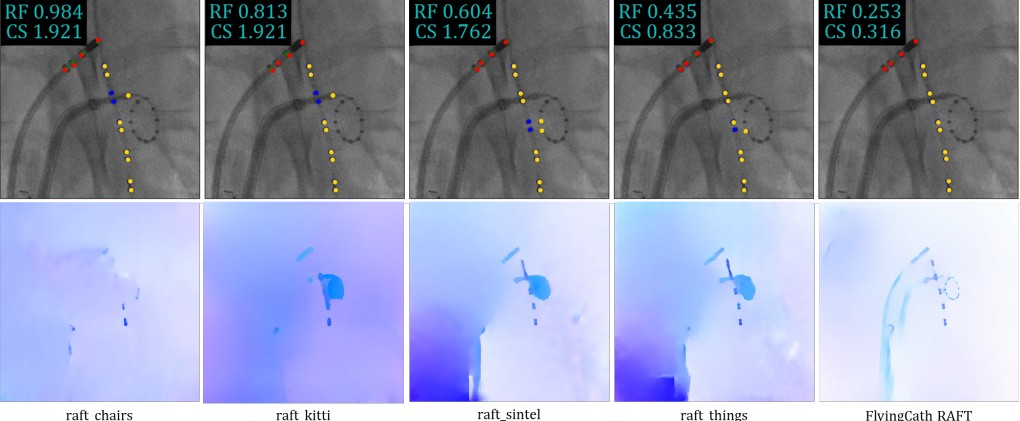

Figure 3: Estimated optical flow map of the same frame of four RAFTs. RAFT trained on Flying-Catheter presents clearer delineation of the catheters and electrodes.

Table 2: Abalation on the additional information. **Best** and Second Best are highlighted.

| | | Test Dataset | | | | | |
|---|---|---|---|---|---|---|---|
| | **RF Catheter** | | | | **CS Catheter** | | |
| **Information** | MRE±STD↓ | SDR (%)↑ | | | MRE±STD ↓ | SDR (%)↑ | | |
| | (mm) | 1mm | 2mm | 4mm | (mm) | 1mm | 2mm | 4mm |
| SAM | 1.12±2.57 | 89.88 | 94.51 | 95.09 | 0.82±1.81 | 93.93 | 97.69 | 98.55 |
| Subsequent Frame | 1.32±3.13 | 85.26 | 93.35 | 95.38 | 1.02±4.64 | 91.33 | 97.11 | 97.98 |
| Original RAFT | 1.93±6.25 | 87.57 | 90.17 | 91.62 | 1.22±2.19 | 89.60 | 94.22 | 97.69 |
| OFELIA (w/o) extra info. | 2.03±5.00 | 81.79 | 86.42 | 89.60 | 1.38±4.75 | 80.92 | 89.60 | 95.38 |
| OFELIA (Ours) | **0.95±2.02** | **90.17** | **95.38** | **96.82** | **0.71±1.75** | **95.66** | **98.27** | **99.42** |

## 5. Conclusion and Future Work

Accurate and efficient electrode detection in real-time fluoroscopy holds a paramount significance. In this work, we propose OFELIA, which introduces optical flow features to the pipeline for precise electrode localization in X-ray series. To improve the model's utility and generalizability, we propose a saturation-based optical flow dataset construction algorithm and fine-tune the optical flow estimator on the synthetic dataset. The results on the test set and two CCA subsets illustrate the efficiency of our proposed OFELIA compared with several SOTA methods. It's worth noting that this approach may not be limited solely to catheter ablation but can be generalized to other tasks such as motion object detection. In terms of electrode landmark detection tasks, further research could be conducted on the usage of different combinations of loss functions, and the exploration of one-shot or few-shot methods to alleviate the burden of electrode annotation.

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

# Appendix A. Visualization of Samples

## A.1. Visualization of Electrode Landmark Order

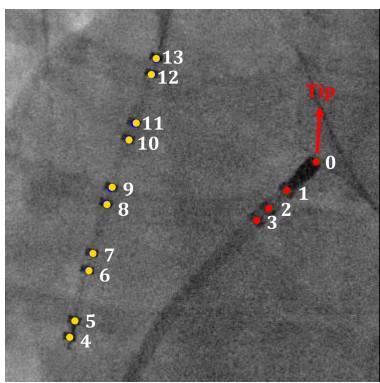

Figure 4: Visualization of Electrode Landmark Order.

## A.2. Samples from Test-DSA Subset

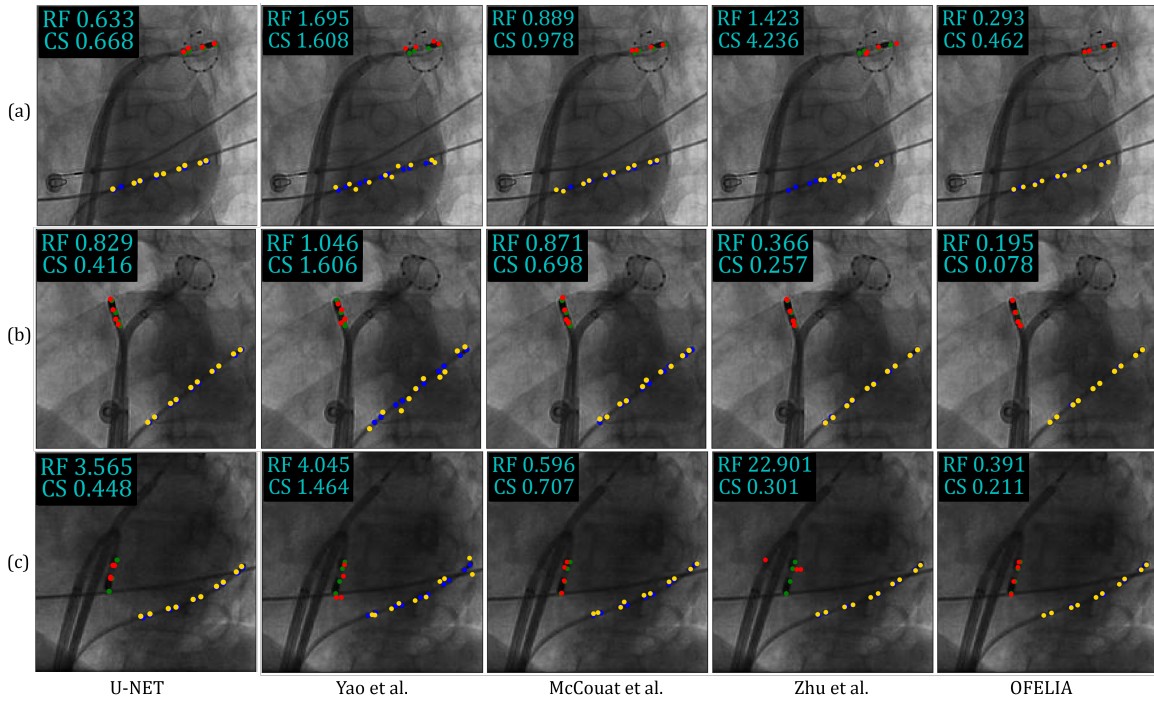

Figure 5: Qualitative results on the Test-DSA Subset. The ground truth and predicted landmark of CS Catheter are in Blue and Yellow. The ground truth and predicted landmark of RF Catheter are in Green and Red.

### A.3. Samples from Test-OBS Subset

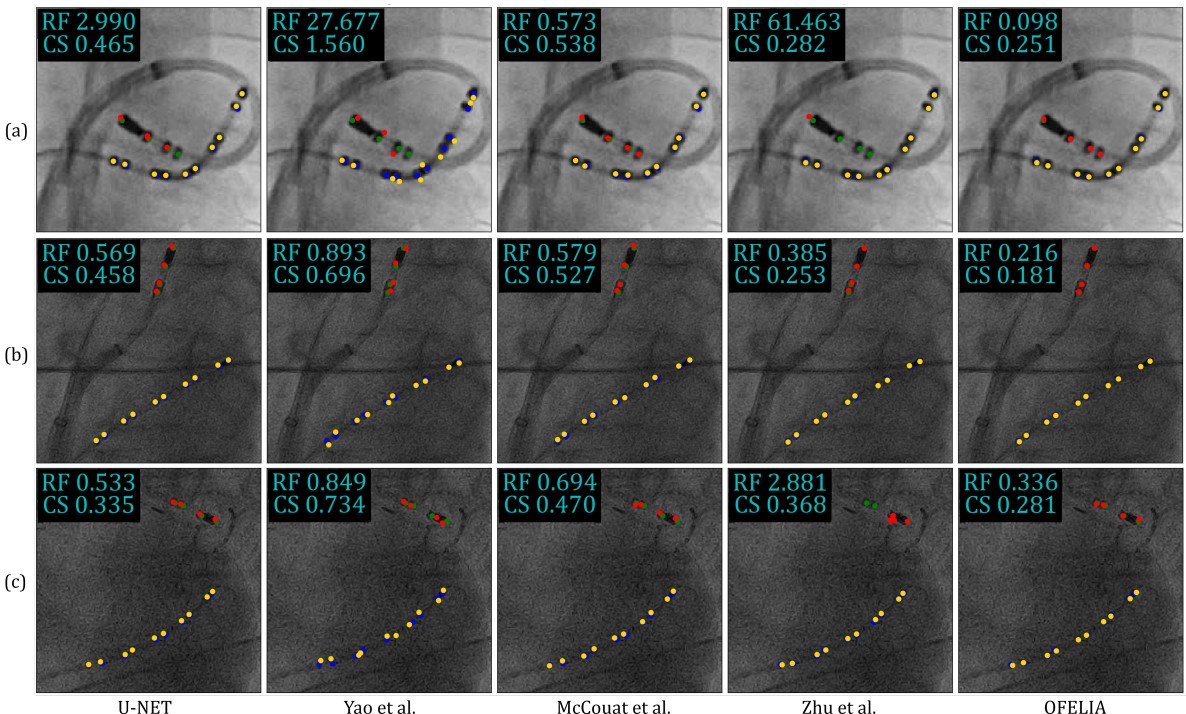

Figure 6: Qualitative results on the Test-OBS Subset. The ground truth and predicted landmark of CS Catheter are in Blue and Yellow. The ground truth and predicted landmark of RF Catheter are in Green and Red.

### A.4. Samples with Part of Landmarks

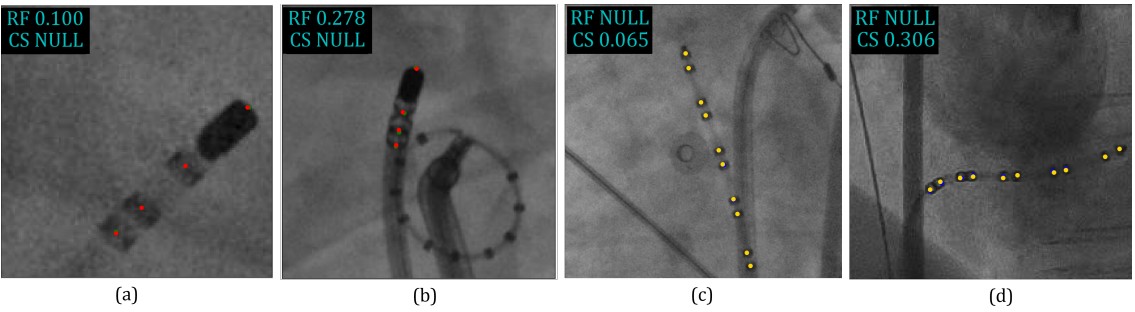

Figure 7: Qualitative results of single catheter cases. The ground truth and predicted landmark of CS Catheter are in Blue and Yellow. The ground truth and predicted landmark of RF Catheter are in Green and Red.

In our study, the highest value in the heatmap can be used to determine whether the predicted landmark is reliable. As shown in Fig 7(a), if the maximum values of the heatmaps corresponding to all electrodes of the CS catheter are less than the conventional threshold, we will conclude that there is no CS catheter in the current frame. This threshold, determined through our statistical analysis, is 30(before normalization).

## Appendix B. Visualization of Failure Detection Cases

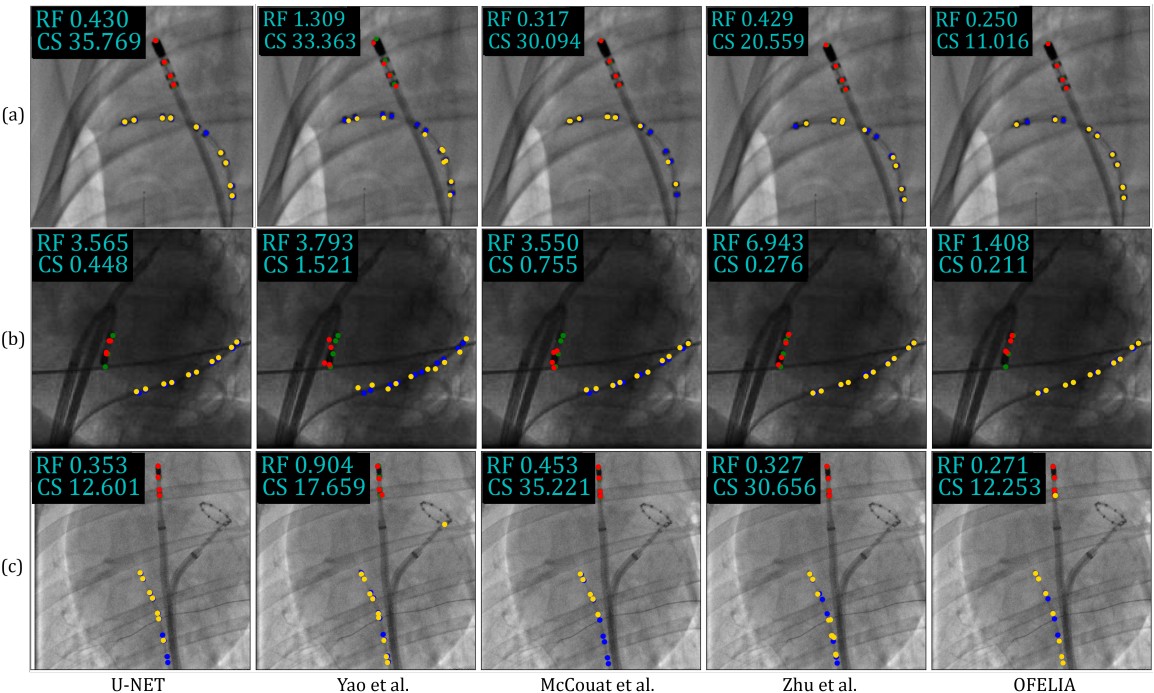

Figure 8: Qualitative results of failure detection cases. The ground truth and predicted landmark of CS Catheter are in Blue and Yellow. The ground truth and predicted landmark of RF Catheter are in Green and Red.

# Appendix C. Statistical Significance Testing of the Results

## C.1. Results on Test Set

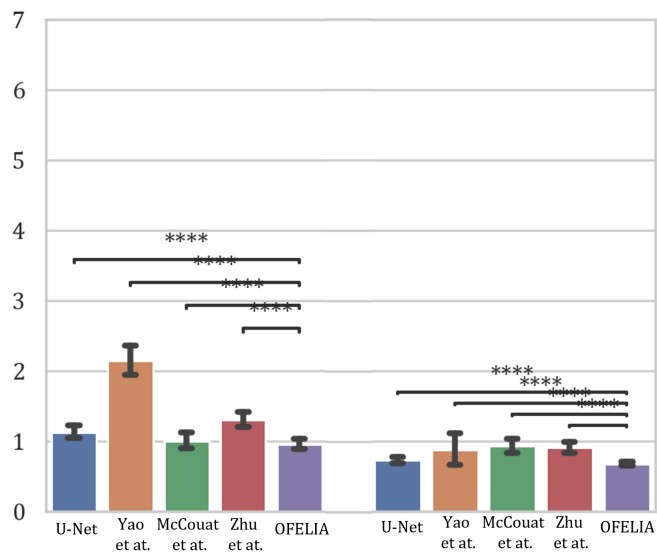

Figure 9: Statical analysis results of test set.

## C.2. Results on Test-DSA Subset

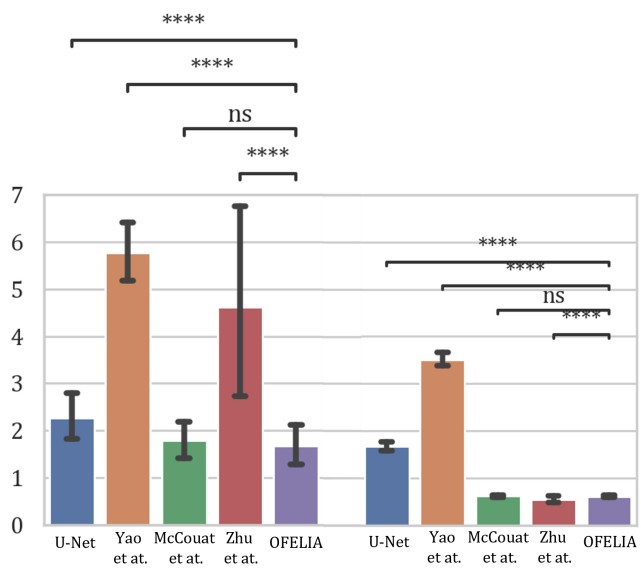

Figure 10: Statical analysis results of Test-DSA Subset.

## C.3. Results on Test-OBS Subset

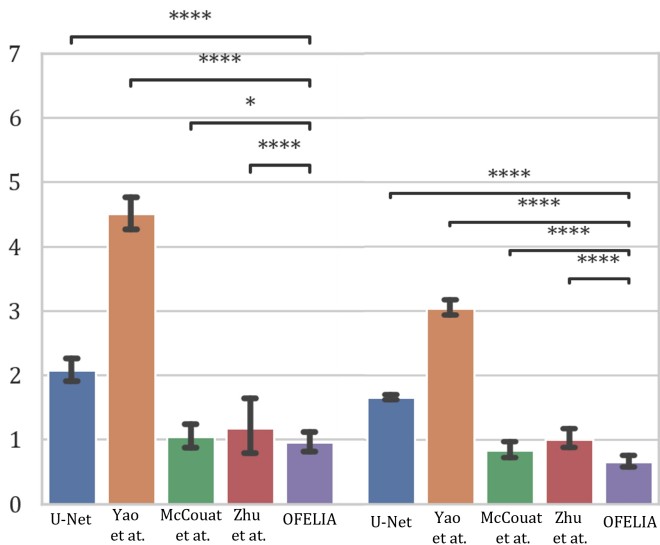

Figure 11: Statical analysis results of Test-OBS Subset.

# Appendix D. Extra Ablation Study

## D.1. Ablation Study on $\alpha$

Table 3: Ablation results on $\alpha$. **Best** and Second Best are highlighted.

| | Test Dataset | | | | | | |
|---|---|---|---|---|---|---|---|
| | **RF Catheter** | | | | **CS Catheter** | | |
| **Model** | MRE±STD ↓ | SDR (%)↑ | | | MRE±STD ↓ | SDR (%)↑ | | |
| | (mm) | 1mm | 2mm | 4mm | (mm) | 1mm | 2mm | 4mm |
| $\alpha = 0$ | 1.93±6.25 | 69.65 | 84.68 | 91.62 | 1.42±7.19 | 88.15 | 93.35 | 97.69 |
| $\alpha = 0.25$ | 1.75± 5.99 | 87.57 | 90.17 | 94.80 | 1.22±4.56 | 89.88 | 94.22 | 98.27 |
| $\alpha = 0.5$(Ours) | **0.95±2.02** | **90.17** | **95.38** | **96.82** | **0.71±1.75** | **95.66** | **98.27** | **99.42** |
| $\alpha = 0.75$ | 3.60±7.57 | 58.38 | 71.39 | 84.39 | 3.04±6.10 | 66.18 | 71.10 | 81.21 |

## D.2. Ablation Study on The Number of Frames Used

Table 4: Ablation results on The Number of Frames Used for RAFT fine-tune Regarding $\alpha$.

| $\alpha$ | 0 | 0.25 | 0.5 | 0.75 |
|---|---|---|---|---|
| Number | 14,768 | 10,345 | 6,212 | 315 |

### D.3. Ablation Study on the Four Optical Flow Estimators

Table 5: Ablation results on the Four Optical Flow Estimators. **Best** and Second Best are highlighted.

| | Test Dataset | | | | | | | |
| --- | --- | --- | --- | --- | --- | --- | --- | --- |
| | RF Catheter | | | | CS Catheter | | | |
| **Model** | MRE±STD ↓ | SDR (%)↑ | | | MRE±STD ↓ | SDR (%)↑ | | |
| | (mm) | 1mm | 2mm | 4mm | (mm) | 1mm | 2mm | 4mm |
| RAFT Chairs | 2.22±5.41 | 81.79 | 85.55 | 87.57 | 0.90±2.20 | 86.99 | 93.35 | 96.24 |
| RAFT kitti | 2.40±7.08 | 75.72 | 88.73 | 91.62 | 2.68±6.34 | 77.46 | 90.17 | 93.93 |
| RAFT things | 2.66±9.37 | 78.61 | 87.57 | 89.60 | 2.21±8.24 | 77.75 | 85.84 | 93.35 |
| RAFT sintel | 1.93±6.25 | 87.57 | 90.17 | 91.62 | 1.42±7.19 | 88.15 | 93.35 | 97.69 |
| OFELIA (Ours) | **0.95±2.02** | **90.17** | **95.38** | **96.82** | **0.71±1.75** | **95.66** | **98.27** | **99.42** |

### D.4. Ablation Study on Using Longer Frame Stacks

Table 6: Ablation results on Using Longer Frame Stacks. Frame123→2 means that we take frame $X_{t-1}, X_t$ and $X_{t+1}$ as input to predict landmark positions on frame $X_t$. The meanings of Frame12→1, Frame1234→2 and Frame12345→3 follow the similar manner. **Best** and Second Best are highlighted.

| | Test Dataset | | | | | | | |
| --- | --- | --- | --- | --- | --- | --- | --- | --- |
| | RF Catheter | | | | CS Catheter | | | |
| **Model** | MRE±STD ↓ | SDR (%)↑ | | | MRE±STD ↓ | SDR (%)↑ | | |
| | (mm) | 1mm | 2mm | 4mm | (mm) | 1mm | 2mm | 4mm |
| Frame12→ 1 | 2.67±4.39 | 81.21 | 86.13 | 89.31 | 1.88±2.96 | 86.42 | 92.49 | 96.82 |
| Frame123→ 2 | 4.24±8.84 | 75.72 | 84.68 | 88.73 | 2.21±3.06 | 69.08 | 78.90 | 87.28 |
| Frame1234→ 2 | 4.33±5.80 | 68.79 | 74.86 | 79.48 | 3.06±5.54 | 77.46 | 85.55 | 91.62 |
| Frame12345→ 3 | 3.15±4.60 | 70.52 | 82.08 | 87.86 | 2.50±5.13 | 71.39 | 76.59 | 87.28 |
| OFELIA (Ours) | **0.95±2.02** | **90.17** | **95.38** | **96.82** | **0.71±1.75** | **95.66** | **98.27** | **99.42** |

### D.5. Ablation Study on Using a Held-Out Center

Table 7: Ablation results on Using a Held-Out Center. To enhance the evaluation, we select one center from our multi-center dataset as a held-out center. The data from this center is not used for training but exclusively used for testing purposes. Thus, we have established a new test set, which we have named as the Test-Plus Dataset, containing 360 sequences.**Best** and Second Best are highlighted.

| | Test-Plus Dataset | | | | | | | |
| --- | --- | --- | --- | --- | --- | --- | --- | --- |
| | RF Catheter | | | | CS Catheter | | | |
| **Model** | MRE±STD ↓ | SDR (%)↑ | | | MRE±STD ↓ | SDR (%)↑ | | |
| | (mm) | 1mm | 2mm | 4mm | (mm) | 1mm | 2mm | 4mm |
| U-Net[*] | 2.86±6.63 | 71.67 | 80.00 | 86.67 | 1.60±4.94 | 78.61 | 87.22 | 93.33 |
| Yao et al.[*] | 5.95±11.37 | 61.67 | 67.22 | 74.17 | 1.16±1.53 | 70.83 | 83.89 | 96.67 |
| McCouat et al.[*] | 2.35±11.25 | 86.67 | 91.67 | 93.33 | 0.91±5.68 | 91.67 | 95.83 | 97.50 |
| Zhu et al.[*] | 1.45±3.55 | 83.61 | 88.61 | 91.94 | 0.83±**1.46** | 87.22 | 93.33 | 95.28 |
| OFELIA (Ours) | **1.18±2.82** | **91.67** | **93.33** | **95.00** | **0.73**±1.53 | **92.50** | **96.67** | **98.33** |

## Appendix E. Visualization of Optical Flows with Different $\alpha$ and Corresponding Catheter Segmentation Maps

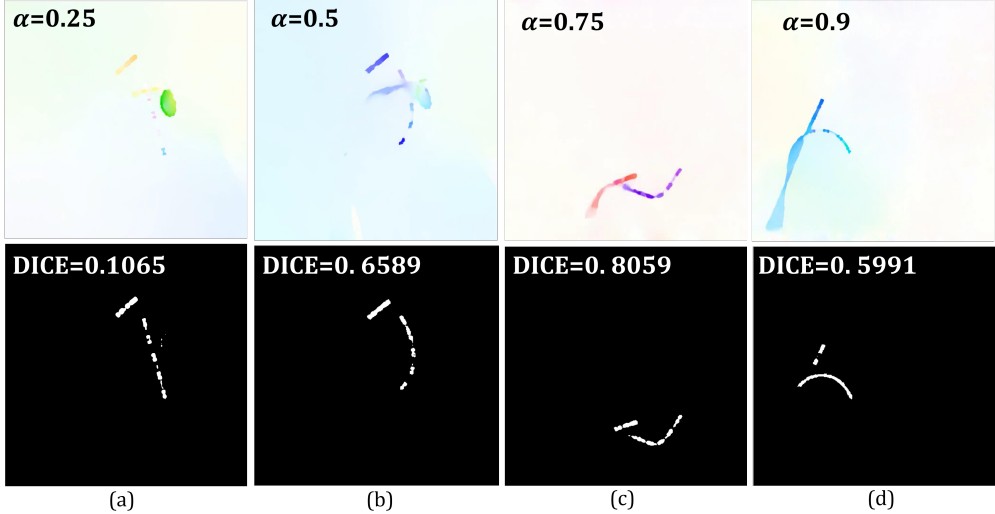

Figure 12: Visualization of Optical Flows with Different $\alpha$. The segmentation mask is generated by SAM (Kirillov et al., 2023) with landmark prompt input.

In our study, we don't have the segmentation masks of catheters, but to better illustrate the $\alpha$ issure, we conduct the ablation study of visualization of optical flows with different $\alpha$ and corresponding catheter segmentation generated by SAM (Kirillov et al., 2023). As it shown in Fig 12, the optical flow maps selected by $\alpha \geq 0.5$ show similarities with the segmentation maps (Dice $\geq 0.5$).

## Appendix F. Details about Dataset Constructions

Two clinical-challenging (CCA) subsets are manually selected from entire test set by clinical experts, like ConTrack (Demoustier et al., 2023), they split their private test dataset into several types according to different scenarios. In our study, the Test-DSA Subset encompasses specific clinical scenarios: X-rays under angiography, where the injection of contrast agents leads to non-uniformed dark shadows moving with the bloodstream in the X-rays, potentially affecting the field of vision. The Test-OBS Subset includes situations in X-rays where catheters obscure each other or are obscured by external wires, patches, etc.. Both of the two subsets present augmented complexity for the detection of catheter electrodes.

## Appendix G. Details about RAFT Fine-tuning

When tackling the issue of video landmark detection, to better observe video quality, we store the optical flow maps in RGB video format, following the conversion process provided by RAFT (Teed and Deng, 2020). During our analysis, we observe that the optical flow map, especially its saturation channel, provides shape information of catheter electrode to

some extent. This saturation map was derived by converting the optical flow images in RGB into the HSV format. Our analysis further reveals that sufficient shape information from optical flow images was accessible when $\alpha$ is greater than 0.5. However, an excessively high $\alpha$ value led to a drastic reduction in the amount of data available for finetuning($\alpha$=0.5 6212 frames, $\alpha$=0.75 315 frames), while too low an $\alpha$ would introduce noisy data. Therefore, we establish the flying catheter dataset with 0.5 as the threshold value and perform finetuning based on *RAFT sintel* model because the *RAFT sintel* model reveals a higher mean SF compared with other three raft models.

