# OpenReview forum: "OFELIA: Optical Flow-based Electrode LocalIzAtion"
_MIDL.io/2024/Conference — MIDL 2024 Poster_

### Official Review · Reviewer_sc8B · 2024-02-28

**Confidence:** 4
**Preliminary Rating:** 3
**Recommendation:** Poster
**Final Rating:** 3.5

**Summary:**

This work proposes a method for locating and tracking ablation catheters in real-time fluoroscopy. The main novelty (in my opinion) is a method to select training data for fine-tuning the RAFT optical flow method for this kind of data, so that it can serve as additional input for the catheter localization model. The latter is modeled as landmark prediction task, with a fixed number of landmarks encoded in separate output channels of U-net like CNN (with pretrained ResNet-34 as encoder). The dataset employed is an in-house dataset.

**Strengths:**

The suggested approach is relatively clearly described and illustrated. The results look quite good. Three prior works are compared against. Two ablation studies shed some light on selected aspects (qualitatively the optical flow, and quantitatively its contribution as additional input to the suggested method).

**Weaknesses:**

The motivation looks a little off to me; AFAIK, it is not the case that such a method is necessary to help humans because "it is difficult for clinicians to localize catheters in real-time fluoroscopy". The merit of such a method should rather be that the resulting geometry can be used for interventional support systems based on models from prior imaging, planning, and 3D models. I am fairly sure that doctors would not perform such procedures frequently if they could not see the catheters.

The prior works used for comparison are probably(?) implemented by the authors themselves (this is not stated explicitly, however). So it is not entirely clear how reliable that part of the evaluation is.

The in-house dataset is used for training and evaluation, and the derived "Flying Catheters" dataset is therefore also not published.  Although it is a multi-center dataset, from the description of the method I understand that it can only work if there are exactly the two catheters (of RF / CS type) with the given 14 landmarks (4 for RF and 10 for CS) visible in each processed frame.

**Detailed Comments:**

I think the paper does not strictly adhere to the page limit?!

The description of the Flying-Catheter dataset is strange; in particular, "the predicted optical flow map is converted to HSV" sounds as if the vector field was an RGB image. (It is usually displayed as such, but the underlying data should be fully sufficient.) Hence, I also question the use of the phrase "saturation-based selection algorithm" – what exactly is the "saturation" of the optical flow? I guess you mean the *magnitude* of the vector field, but that's also not reflected in the formulas (which also do not seem to mention the "saturation" part).  Or maybe I am getting that wrong because I do not fully comprehend why one should strive for maximum mean magnitude over all landmarks as a quality measure for the flow estimation.

In the same way, the meaning of the "quality control threshold α (= 0.5)" is unclear, because the range of the "saturation factor" is unclear.

It is not really well-motivated why all four models had to be used, and it would be interesting to check which model was selected how often.

There are no details on the RAFT finetuning, and it is also not clear which of the four RAFT models was finetuned (probably s, guessing from the ablation study?).

Several spelling mistakes (Abalation, Overivew, Optcial), wrong terms (elastically transformation -> elastic deformation) and formatting (missing spaces before parentheses) should be fixed. The phrase "extract the tip detection results" in related work is not good – what does "extract" refer to? What kind of "tip"? ("the" indicates that the reader should know.)

Formula (2) is superfluous (standard CE loss averaging, AFAICS).

The statement "Combining X-ray images and optical flow maps can drive the neural network to pay more attention to the electrode part of the field of view" is not justified, in particular not in the methods section, before you do any experiments. But I also cannot see which experiment would justify that statement in this form.

Why did you select the sintel model for the ablation study?

**Justification Of Final Rating:**

First of all, I want to highlight that the authors really took time to discuss with the reviewers and to revise their paper, which is great! Maybe the replies would have been more readable if they would have cited the respective review parts directly. Unfortunately, I still wonder what the fire symbol in Figure 1 is about – is it an easter egg? Probably, my (repeated) questions in OpenReview to this point were not clear enough due to the way they got displayed. But this is not a critical point I think.

To conclude, the paper was revised and I will raise my rating, but only very slightly because my overall impression of the work did not fundamentally change.

**Justification Of The Preliminary Rating:**

The work is potentially interesting, the illustrated results look promising and the paper is readable, but many technical details are missing and the evaluation is not fully reliable and trustworthy in the light of the above questions and weaknesses.

**Questions To Address In The Rebuttal:**

Have you performed statistical significance tests when determining the "best" and "second best" values in the tables?

The description of the CCA test sets is incomplete:
1) Does the training data also contain such CCA frames?
2) Are the three test sets "Test", "DSA-Test" and "OBS-Test" disjoint? Or what are their (subset) relations?

Are the radii Euclidean distances or measured orthogonally to the catheter?

Does the training procedure really only have a single 1 in every output channel? No smoothing? How is the preprocessing (geometric transformations) implemented in order not to lose that 1?

Where is "This attention-based mechanism" that you refer to?

**Special Issue:**

No

---

> ### Author Response · Authors · 2024-03-17
> **Responds to Reviewer sc8B**
>
> Thanks for your review, which greatly enhances the completeness of our work and experiments.
>
> Respond to Weakness: We have modified the length of the paper and the main motivation of our method, the future applications based on electrode localization are surely of concern to us. As you mentioned, if doctors cannot see the catheter, they would not perform such surgery. However, in some complex intraoperative scenarios, such as contrast agent injections and catheter intersections during cardiac ablation procedure, doctors may also experience discontinuity or difficulty in observing electrode positions. Hence, this is why we extract these two kinds of special scenario cases as the CCA dataset.
>
> The prior works are evaluated using the official code and we will add footnotes in the result table. We have added visualizations where only part of the landmarks are visible in the images, our OFELIA can still tackle this problem correctly under this condition.
>
> Respond to Comments:
> When tackling the issue of video landmark detection, to better observe video quality, we store the optical flow maps in RGB video format, following the conversion process provided by RAFT. During our analysis,  we observe that the optical flow map, especially its saturation channel,  provides shape and boundary information of catheter electrode to some extent.
> This saturation map is derived by converting the optical flow images in RGB into the HSV format. Our analysis further reveals that these shape information from optical flow images are accessible when alpha was greater than 0.5. However, an excessively high alpha value lead to a drastic reduction in the amount of data available for finetuning($\alpha$=0.5 6212 frames, $\alpha$=0.75 315 frames), while too low an alpha would introduce noisy data. Therefore, we establish the flying catheter dataset with 0.5 as the threshold and perform finetuning based on raft_sintel model because the raft_sintel model reveals a higher mean SF compared with other three raft models.
>
> We have corrected the typos in the manuscript. To explain the order of landmarks and the position of the tip, we add a schematic figure of the catheter structure in the appendix A.1.
>
> Our loss is computed by averaging $L_{CE}$ of each landmark instead of the difference of the whole output, please see R1W3 for details.
>
> As shown in the appendix E, the optical flow map generated by our method($\alpha=0.5$) shows similarities with the SAM generated segmentation maps (Dice > 0.5). And concatenating the image and flow map before sent to the U-Net will make the network focus more on these areas.
>
> We conduct an ablation study on all four pre-trained optical flow estimators in the appendix D.3, and sintel model reveals better MRE compare to the others.
>
> Respond to Q1: We have added statistical significance tests in the appendix C., our OFELIA outperforms other methods with a significance of p < 0.01.
>
> Respond to Q2: The training data contains a total of 560 sequences(14768 frames), within the training dataset, we have 61 DSA sequences (827 frames) and 174 OBS sequences (4342 frames). Test-DSA Subset contains 53 sequences (575 frames) and Test-OBS Subset 145 sequences ( 2266 frames), they are disjoint test sets manually split by experienced clinicians, please refer to R2C4 and the appendix F for the detailed dataset construction method.
>
> Respond to Q3: We calculate the Euclidean distances between landmarks and their corresponding groundtruth.
>
> Respond to Q4: Yes, we do not conduct smoothing. Even without smoothing we can train successfully.
>
> Respond to Q5: Sorry for confusion, we now delete the phrase.

---

### Official Review · Reviewer_62n7 · 2024-03-04

**Confidence:** 3
**Preliminary Rating:** 3
**Recommendation:** Poster
**Final Rating:** 3.5

**Summary:**

The authors propose a deep neural networks-based approach for the localization of catheter electrodes in fluoroscopic x-ray sequences. The proposed approach integrates temporal information in each frame via optical flow maps obtained from the previous frame in the sequence. To obtain high-quality optical flow maps for training, outputs of four pre-trained models for optical flow estimation in natural images are used to produce refined realistic optical flow maps, which are then used as ground truths for the optical flow map estimation network. The proposed approach outperforms state-of-the-art localization networks.

**Strengths:**

The paper is well-written and easy to follow. The proposed idea is also simple and effective.
The authors do not just directly adopt optical flow estimation networks trained on natural images, they also refine the maps obtained from four pre-trained networks and use them as ground truths for optical flow estimation networks. I think this is a very strong point, which is generally ignored in the medical imaging community.
The proposed idea leads to an annotation-efficient pipeline using landmarks as opposed to segmentation masks, which can be time-consuming.
The proposed approach outperforms state-of-the-art methods, and the ablation study is also well-performed.

**Weaknesses:**

Although the paper is well written, there are some missing details both in terms of results and methodologies, which I am listing in the 'detailed comments' and 'questions to be addressed in the rebuttal'. Furthermore, the proposed approach is demonstrated on a single dataset. It is difficult to tell how generalizable the proposed approach is.

**Detailed Comments:**

1 - In the introduction section where contributions are mentioned, point number 1 already encapsulates 3. The temporal information is integrated through optical flow. I would suggest the authors to merge them.
2 - The authors mention multiple times that annotation of segmentation masks is more time-consuming compared to annotating landmarks. How much is the annotation time difference per sequence (approximately)?
3 - In the dataset section, the authors list the number of frames used during training and testing. How many sequences were there? Were there overlaps between training and testing sequences?
4 - How were the clinically challenging subsets (CCA) extracted?
5 - In the implementation details section, how were the hyperparameters selected considering there is no validation data? Especially how was alpha selected? What is the effect of alpha on the final performance?
6 - Figure 2 should also contain samples from CCA subsets.
7 - In the tables the mean values of MRE and SDR are noted. These values can be sometimes misleading. Please also list the standard errors across sequences or show bar plots showing distributions of all methods, and perform statistical significance testing. There are some types in the tables (e.g second best method in table 2 is SAM in columns 1 and 3)

**Justification Of Final Rating:**

I believe most of my concerns were addressed. However, I am not completely satisfied with the hyperparameter selection procedure, and the held-out site experiments were not performed for the competing methods.
Overall, I'm making it a borderline accept.

**Justification Of The Preliminary Rating:**

I believe that the proposed approach is promising, however, some issues need to be addressed. These issues include fewer details in the methodology and results section and limited demonstration on a single dataset. I am willing to accept if they are addressed.

**Questions To Address In The Rebuttal:**

1 - In addition to the mean values across sequences (in tables), the metrics should also include information regarding the distribution of the metrics in the form of standard error or bar plots. Furthermore, statistical significance tests should also be performed.
2 - Please add more figures (like figure 2) in the appendix/supplementary materials, and also show failure cases and some cases from the CCA subset.
3 - Please explain the criterion to select CCA subset, and how was the distribution of CCA subsets in the training and testing splits.
4 - Please explain how alpha was selected, and how having SF above a specific threshold can help in producing better optical flow maps. What is the effect of alpha on the optical flow estimation networks?
5 - The proposed approach is demonstrated only on a single dataset (although it is multi-center). How would the proposed approach perform on other datasets? Did the author perform any held-out center experiments? It is crucial to evaluate the effectiveness of the proposed approach.

**Special Issue:**

No

---

> ### Author Response · Authors · 2024-03-17
> **Responds to Reviewer 62n7**
>
> Thanks for your review, which greatly enhances the completeness of our work and experiments.
>
> Respond to Comment 1: We have merged the contributions in the revised version.
>
> Respond to Comment 2: It takes the engineers about 15 min to annotate the landmarks of an X-ray video with 100 frames, while annotating the segmentation mask requires nearly 1.5h, about 5 times longer.
>
> Respond to Comment 3: There are 560 sequences (14,768 frames) for training 346 sequences (7,711) test set, without any overlap. Besides, there are 53 sequences (575 frames) Test-DSA Subset, and 145 sequences (2,266 frames) Test-OBS Subset are seected from test set, there is no overlap between DSA subset  and OBS subset.
>
> Respond to Comment 4: Two CCA subsets are manually selected from entire test set by clinical experts, like ConTrack, they split their private test dataset into several types according to different scenarios. In our study, the DSA Set encompasses specific clinical scenarios: X-rays under angiography, where the injection of contrast agents leads to non-uniformed dark shadows moving with the bloodstream in the X-rays, potentially affecting the field of vision. The OBS Set includes situations in X-rays where catheters obscure each other or are obscured by external wires, patches, etc.. Both of the two subsets present augmented complexity for the detection of catheter electrodes.
>
> Respond to Comment 5: We have added an ablation study on $\alpha$ in the appendix D.1. From the table, we can find that $\alpha=0.5$ has the best performance. This is because a high $\alpha$ gives strict quality control on the Fly-Catheter Dataset, resulting in an insufficient number of train samples. A low $\alpha$ introduces too many noisy samples to the dataset, thus hurting the final utility.
>
> Respond to Comment 6: Figure 2(b) is a sample from DSA subset and Figure 2(c) is from OBS subset, we have added more caption under Figure2. To further address this issue, we have added more samples from CCA subsets in the appendix A.2 and appendix A.3.
>
> Respond to Comment 7: We have added standard errors in the Tables, and added bar plots in the appendixes C.1, C.2 and C.3.
>
> Respond to Q1: We have added standard errors in the Tables, and added bar plots in the appendixes C.1, C.2 and C.3..
>
> Respond to Q2: We have added figures showing failure cases and cases from CCA, please see the appendix B.
>
> Respond to Q3: Please refer to Respond to Comment 4.
>
> Respond to Q4: Please refer to Respond to Comment 5, we also show 5 optical flows with $\alpha=0.25, \alpha=0.5, \alpha=0.75, \alpha=0.9$ in the appendix E. As shown in the Figure, optical flow maps with low $\alpha$ cannot provide enough accurate shape information for landmark detection, and a high $\alpha$ leads to a rapid decrease of the number of samples ($\alpha$=0.25 10345 frames, $\alpha$=0.5 6212 frames, $\alpha$=0.75 315 frames) for estimator finetuning.
>
> Respond to Q5: As the electrode landmark detection in catheter ablation is a less popular but important direction, to the best of our knowledge, there is no publicly available dataset. However, we have added a simple baseline – the origin U-Net to the result Table 1 for better understanding of the difficulty of this task.

---

> > ### Comment · Reviewer_62n7 · 2024-03-26
> >
> > Thank you for addressing most of the comments. I have a few questions:
> > 1 - How were hyperparameters selected? Is there validation data?
> > 2 - The data is multi-centered. Do you have information regarding each center that could be used as a held-out center(during training) and only used during testing?
> > 3 - Are standard errors and statistical significance tests across frames or sequences?

---

> > > ### Author Response · Authors · 2024-03-27
> > > **New Respond to Reviewer 62n7**
> > >
> > > Thank you for your review.
> > >
> > > Respond to Question 1: We train for 20 epochs, saving both the final model and the best model in terms of training loss, then evaluate the performance using the best model.
> > >
> > > Respond to Question 2: We mix all the data from multicenters and randomly divide the mixed data into training and testing. There is no heldout center in our study. For this question, we conduct a held-out test. We select one group from the multicenter data solely as the test set and re-train on the remaining training set. Due to time constraints, we only train OFELIA. The results are as follows: RF Catheter: 0.95$\pm$3.95 93.42 96.41 97.40, CS Catheter: 1.31$\pm$2.43 94.99 96.90 98.01. These new results do not differ significantly from original OFELIA in Table 1.
> > >
> > > Respond to Question 3: The standard errors and statistical significance tests are evaluated across frames.
> > >
> > > I hope these answers address your questions.

---

> ### Comment · Reviewer_62n7 · 2024-03-27
>
> Thank you again.
> About the answers to my questions:
> 1 - It's not a good idea to use training loss to select hyperparameters
> 2 - Please add them to the final version of the main manuscript, and If possible please also calculate the results for the competing methods.
> 3 - Please calculate across sequences not frames.
> I'll be changing my decision to borderline accept. However, I trust that you will add 2 and 3 to the final version of the manuscript.

---

> > ### Author Response · Authors · 2024-03-28
> > **New Respond to Reviewer 62n7**
> >
> > Thank you for your review, which greatly enhances the completeness of our work and experiments. I will run the experiments based on your suggestions and make modifications in the main manuscript accordingly.

---

### Official Review · Reviewer_5zr1 · 2024-03-05

**Confidence:** 4
**Preliminary Rating:** 2
**Final Rating:** 3.5

**Summary:**

This paper presents an approach for the localisation of electrodes over a series of temporal X-ray images. The authors propose using optical flow over two frames to gather temporal information which is then used to predict the locations of electrodes in the first of the two images. They demonstrate results on an in-house dataset, where the proposed approach performs better than three competing ones. Two subsets of the dataset, which consist of particularly difficult examples, are also evaluated and the proposed method is shown to be better in terms of two metrics. The paper also proposed an algorithm to construct a dataset on-the-fly to improve optical flow on the type of image observed in the task.

**Strengths:**

1. The paper adds optical flow-based landmark localisation for detection of electrodes from a sequence of X-ray images.
2. An algorithm to construct a dataset on-the-fly is proposed, which is able to adapt a pre-trained optical model to the type of X-ray image observed in the task.

**Weaknesses:**

1. The application of optical flow seems to be incomplete. The paper claims that it views the problem as a video landmark detection network, yet optical flow is used only for two consecutive frames, with the resulting localisation information not being used for further detections. The authors could explore using longer frame stacks, adding smoothing constraints to landmark locations over time, etc. As it is presented, the methodology seems to lack several possible explorations.
2. The authors mention Demoustier et al. 2023 as the closest work to their paper, yet it is not compared/evaluated against the proposed version. Indeed, Demoustier et al. 2023 also propose using temporal context to refine detections.
3. It is unclear if the index of each landmark has semantic identity. For example, in Figure 2, what is the semantic difference between landmarks appearing on the two catheters? This is important because the training loss optimises for the detection of each landmark separately, and there is no consensus between these detections (i.e., each landmark is detected individually).
4. The paper claims to evaluate the approach on three datasets, but it is the same dataset with two additional subsets from it.

**Detailed Comments:**

Overall, the paper seems incomplete in its exploration of possibilities. The formulation for the loss is not well motivated, as the semantic meaning of landmarks (for example, why is landmark number 2 always landmark number 2? Is it guaranteed?) is unclear. There are not enough images shown in the paper to explain the dataset and for qualitative analysis of the results. In my opinion, the proposed approach can also be evaluated on datasets used in Zhu et al. 2022, as the given approach is not necessarily singular to cardiac ablation procedures.

**Justification Of Final Rating:**

I am changing my rating to borderline accept as the authors have answered questions raised in the rebuttal well. However, I would encourage the authors to add a short discussion paragraph on how this approach is not specific to catheter ablation but can be generalised to other types of data too.

**Justification Of The Preliminary Rating:**

In my opinion, the paper is incomplete and not ready for publication at this stage. Several benchmarking and baseline experiments are missing, as noted above. The writing should also be improved as several points are unclear.

**Questions To Address In The Rebuttal:**

I would appreciate if the authors could clarify some minor things.

1. In algorithm 1, F^hat is converted to the HSV colour space. What does this mean? Does this refer to the visualisation of the optical flow using the HSV colour space? If so, the saturation in this visualisation is set to 255, meaning that the saturation factor computed in Equation 3 will always be the same. Furthermore, what is the importance of the HSV colour space? This needs to be motivated better.
2. There is a missing comparison with a simple baseline U-Net which regresses landmarks from X_t. As the dataset used is in-house, establishing such a baseline is important in order to demonstrate to what extent simple and straightforward approaches work on it.
3. What is the purpose of training \phi* if it is not used in Algorithm 1? Are the demonstrated results obtained with \phi* or with \phi? If yes, is there a comparison with the other (i.e., if results are with \phi, are there results with \phi*, and vice versa)? Also, why is image X_{t+1} not included in D_{f-c} as it will also be needed to fine-tune the optical flow network?

---

> ### Author Response · Authors · 2024-03-17
> **Responds to Reviewer 5zr1**
>
> Thanks for your review, which greatly enhances the completeness of our work and experiments.
>
> Respond to Weakness 1: We have added an ablation study on using longer frame stacks in appendix D.4.
>
> Respond to Weakness 2: We are unable to compare with ConTrack because we do not have segmentation masks.
>
> Respond to Weakness 3: The landmarks in our dataset are arranged in a specific order, with all landmarks positioned at the center of the electrodes, except for the first landmark, which is located at the tip of the RF catheter. To better illustrate the positions and order of landmarks, we have chosen an image annotated with the positions of the landmarks and their corresponding order, which is included in the appendix A.1.
>
> Respond to Weakness 4: Two CCA subsets are manually selected from entire test set by clinical experts, like ConTrack, they split their private test dataset into several types according to different scenarios. We now change the phrases Test-DSA Set and Test-OBS Set to Test-DSA Subset and Test-OBS Subset to avoid confusion.
>
> Respond to Question 1: When tackling the issue of video landmark detection, to better observe video quality, we store the optical flow maps in RGB video format, following the conversion process provided by RAFT. During our analysis,  we observe that the optical flow map, especially its saturation channel,  provides shape and boundary information of catheter electrode to some extent. This saturation map is derived by converting the optical flow images in RGB into the HSV format. Our analysis further reveals that these shape information from optical flow images are accessible when alpha was greater than 0.5. However, an excessively high alpha value lead to a drastic reduction in the amount of data available for finetuning($\alpha$=0.5 6212 frames, $\alpha$=0.75 315 frames), while too low an alpha would introduce noisy data. Therefore, we establish the flying catheter dataset with 0.5 as the threshold and perform finetuning based on raft_sintel model because the raft_sintel model reveals a higher mean SF compared with other three raft models.
>
> Respond to Question 2: Thanks for your suggestion, we now add a baseline U-Net in Table 1.
>
> Respond to Question 3: We have corrected the typos in the manuscript.

---

> ### Comment · Reviewer_5zr1 · 2024-03-27
>
> I thank the authors for their response.
>
> 1. It seems taking longer stacks for estimating landmark positions greatly increases error, almost doubles to quadruples it (Table 6 in updated PDF). This seems quite counter-intuitive. Why do you think this is? Also, isn't Frame12->1 in Table 6 the same as OFELIA? What I understood is OFELIA uses frames at t=t, and t=t+1 to predict landmark locations at t=t, which would be the same as Frame12->1. Also, how are multiple frames implemented in the model? Optical flow consumes only two images, so details on how multiple frames were used should be included.
>
> 2. Comparison against ConTrack seems difficult on the in-house dataset because of lack of segmentation masks. However, I would like the authors to comment on my other observation, which was that the proposed algorithm does not seem to be specific to cardiac ablation procedures. Is it possible to include performance on public datasets (Cephalogram, Hand, Chest) from Zhu et al. 2022, Learning to localize cross-anatomy landmarks in x-ray images with a universal model?
>
> 3. It is still not clear, what the reasoning behind choosing the saturation channel in the HSV space is. Optical flow on two (H x W x 3) images is one (H x W x 2) image, so an image with two channels. If RAFT has a conversion process to visualise this as an RGB image, and this image viewed in HSV space is somehow helpful in providing shape and boundary locations, then this part of the methodology is not evident at all.
>
> 4. Could the authors please comment on Question 3 in the original review (what is the use of \phi* when it is never used)?

---

> > ### Author Response · Authors · 2024-03-27
> > **New Respond to Reviewer 5zr1**
> >
> > Thank you for your review.
> >
> > Respond to Question 1: The ablation study on longer stacks uses the X-ray images only, we didn’t incorporate optical flow maps. In X-ray sequences, the catheters occupy only a small portion of the frame most of the time, and the landmarks are concentrated at a specific point within it. Using longer stacks adds spatial information but lacks concentration, potentially introducing noise instead. Incorporating additional constraints or templates might yield better results.
> >
> > Frame12$\rightarrow$ 1 is different from OFELIA, OFELIA takes frame $X_t$ and optical flow map $F_{t\rightarrow t+1}$ generated by $\varphi^{*}$ as input while Frame12$\rightarrow$ 1 takes frame $X_t$ and $X_{t+1}$ as input, $\textbf{without optical flow map}$. The first and second frames of the X-ray sequence often appear very similar, sometimes providing redundant information, but optical flow map detects pixel differences, offering temporal information and boundary information to some extent, which finally leads to a better result.
> >
> > Respond to Question 2: We agree that this approach is not specific to cardiac ablation procedures, and OFELIA could be used for other tasks involving video detection as well. OFELIA uses inter-frame information in sequences by using the optical flow maps generated by fine-tuned RAFT network ,the optical flow maps also contain certain boundary information and reduce noise or redundacy between consecutive frames in the X-ray sequence or other videos.
> >
> > OFILIA requires information between frames, whereas the dataset used by Zhu et al. consists of $\textbf{single images, lacking sequential information}$. We believe that OFELIA can achieve good results in handling problems other than Catheter Ablation. In the future, we will construct datasets with various types of landmark sequences and test the generalization of OFELIA.
> >
> > Respond to Question 3: For the convenience of visualization, utilizing the code provided by RAFT, we can convert an HxWx2 optical flow image into an HxWx3 RGB image based on the standard flow color wheel. During our experiments, we observe background noise when directly visualizing with RGB. We find that converting to the saturation channel $\textbf{helps to suppress this noise}$ and constrain its values within the range of 0 to 1. Therefore, we employ the saturation channel for filtering and finally building the flyingcath dataset.
> >
> > Respond to Question 4: The fine-tuned $\varphi^*$ in Algorithm 1 is used in OFELIA. We have corrected the typos in $\textbf{Method section}$. The $\varphi^*$ is used for the optical flow prediction between frames: $F_{t\rightarrow t+1} = \varphi^*(X_t, X_{t+1})$.
> >
> > We hope these answers address your questions.

---

### Meta-Review · Area_Chair_WMZd · 2024-04-03

**Recommendation:** Accept (Poster)
**Confidence:** 3

**Metareview:**

This work was a true borderline case with many remarks from the reviewers. During the rebuttal phase, authors were able to explain and address many points, such that all reviewers increased their ratings to "borderline accept". This is certainly not a strong acceptance vote (one of the reviewers wrote that his/her "overall impression of the work did not fundamentally change"), but I think it can be acknowledged that all reviewers increased their initial ratings.

---

### Decision · Program_Chairs · 2024-04-05

Accept (Poster)